# Counterfactual Contrastive Learning for Weakly-Supervised Vision-Language Grounding

**Zhu Zhang**[1,2] , **Zhou Zhao**[1,2*]**, Zhijie Lin**[1] **, Jieming Zhu**[3] **, and Xiuqiang He**[3]
[1]Zhejiang University
[2]Key Laboratory Foundation of Information Perception and Systems for Public Security of MIIT,
Nanjing University of Science and Technology
[3]Huawei Noah's Ark Lab
{zhangzhu, zhaozhou, linzhijie}@zju.edu.cn
{jamie.zhu, hexiuqiang1}@huawei.com

## Abstract

Weakly-supervised vision-language grounding aims to localize a target moment in a video or a specific region in an image according to the given sentence query, where only video-level or image-level sentence annotations are provided during training. Most existing approaches employ the MIL-based or reconstruction-based paradigms for the WSVLG task, but the former heavily depends on the quality of randomly-selected negative samples and the latter cannot directly optimize the visual-textual alignment score. In this paper, we propose a novel Counterfactual Contrastive Learning (CCL) to develop sufficient contrastive training between counterfactual positive and negative results, which are based on robust and destructive counterfactual transformations. Concretely, we design three counterfactual transformation strategies from the feature-, interaction- and relation-level, where the feature-level method damages the visual features of selected proposals, interaction-level approach confuses the vision-language interaction and relation-level strategy destroys the context clues in proposal relationships. Extensive experiments on five vision-language grounding datasets verify the effectiveness of our CCL paradigm.

## 1 Introduction

Vision-language grounding is a fundamental and crucial problem in multi-modal understanding. Video grounding [14, 17] aims to identify the temporal boundaries of the target moment according to the given description. And image grounding [20, 28, 46] localizes a specific region described by a referring expression. Recently, to avoid expensive manual annotations, researchers begin to explore Weakly-Supervised Vision-Language Grounding (WSVLG), which only needs the video-sentence or image-sentence pairs during training. Most existing approaches [33, 26, 11, 29, 15, 23, 8, 37] employ the MIL-based or reconstruction-based paradigm to train weakly-supervised grounding networks, but they both have some drawbacks. The MIL-based methods [11, 29, 15, 8] often define the original vision-language pairs as positive samples, construct the unmatched vision-language pairs as negative samples, and directly learn the latent visual-textual alignment by an inter-sample loss. However, these approaches heavily depend on the quality of randomly-selected negative samples, which are often easy to distinguish and cannot provide strong supervision signals. On the other hand, reconstruction-based methods [33, 26, 23, 37] attempt to reconstruct the sentence query from visual contents during training and utilize intermediate results, such as attention weights, to localize the target proposal (i.e., region or moment) during inference. But these methods cannot directly optimize the visual-textual alignment scores which are applied for inference. Considering the proposals with

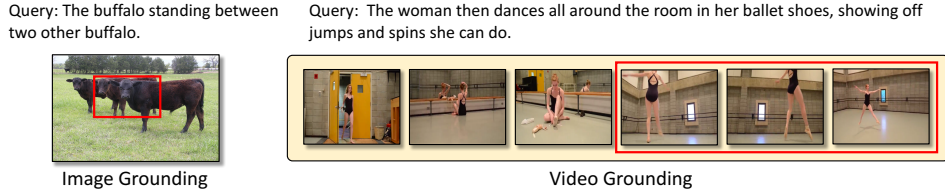

Query: The buffalo standing between two other buffalo.

Query: The woman then dances all around the room in her ballet shoes, showing off jumps and spins she can do.

Image Grounding

Video Grounding

Figure 1: Examples of image grounding and video grounding.

higher weights are not necessarily more relevant to the sentence query [21], the indirect optimization may restrict further performance improvement.

Recently, contrastive learning has greatly promoted unsupervised pretraining of visual representations [16, 7], which develops contrastive training between positive and negative samples to learn expressive visual features. In this paper, we propose a novel Counterfactual Contrastive Learning (CCL) paradigm for the WSVLG task, which constructs fine-grained supervision signals from counterfactual results to directly optimize the visual-textual alignment. Specifically, we first employ a MIL-based pre-trained grounding network to estimate the given vision-language pairs to produce original results. By the gradient-based selection method, we then build a critical proposal set and an inessential proposal set. Next, we design Robust Counterfactual Transformations (RCT) based on the inessential set and devise Destructive Counterfactual Transformations (DCT) according to the critical set. After it, we apply the grounding network with constructed RCT and DCT to generate counterfactual results of the vision-language pairs, including positive and negative results corresponding to RCT and DCT, respectively. Finally, we develop a ranking loss to focus on the score-based difference between positive and negative results, and further devise the consistency loss to consider the distribution-based discrepancy between them. To make the contrastive training effective, the network with DCT needs to generate plausible negative results, where crucial visual contents corresponding to the sentence query are destroyed while unnecessary contents are retained. On the contrary, the network with RCT should damage visual contents of inessential proposals and produce robust positive results. Thus, we design the counterfactual transformations from three levels: (1) the feature-level strategy damages the features (i.e. endogenous clues) of selected proposals by the memory-based replacement; (2) the interaction-level strategy confuses the vision-language interaction by destroying the multi-modal fusion; and (3) the relation-level strategy perturbs the context relations (i.e. exogenous clues) of chosen proposals by counterfactual relation construction. The three strategies are applied to the intermediate process of network inference rather than the raw inputs, and produce ambiguous results for sufficient contrastive learning.

The main contributions of this paper are summarized as follows:

- We propose a novel counterfactual contrastive learning for WSVLG, which develops sufficient contrastive training between counterfactual positive and negative results.

- We design the feature-level, interaction-level and relation-level strategies for counterfactual transformations, which are applied to the intermediate process of network inference.

- Our CCL not only focuses on the score-based difference between the positive and negative results but also considers the distribution-based discrepancy between them.

- We conduct extensive experiments on five large-scale vision-language grounding datasets to verify the effectiveness of our proposed CCL paradigm.

## 2 Related Work

**Image Grounding.** Image grounding aims to localize the object region corresponding to the given referring expression. Early supervised approaches [20, 28, 47, 30, 48] directly learn the latent visual-textual alignment. And further works explore the language expression decomposition [46, 19], co-attention interaction [12, 55] and relation construction [44, 43]. Besides, Liu et al. [27] introduce the adversarial erasing approaches [36, 40] into this field, which detect the crucial contents by the Grad-CAM method [35] and hide these contents to make the network further focus on other relevant contents. Under the weakly-supervised setting, researchers build the supervision signals by the

reconstruction-based paradigm [33, 4, 54, 26] and MIL-based paradigm [51, 54, 11]. Rohrbach et al. [33] learn the intermediate attention weights to localize the region by language reconstruction, and Chen et al. [4] design knowledge-aided consistency in both visual and language modalities. Further, Liu et al. [26] devise the collaborative training of language reconstruction and attribute classification. From the MIL-based view, Zhang et al. [51] propose a variational Bayesian method to explore the context modeling and Datta et al. [11] build caption-conditioned image encoding to infer the latent region-phrase correspondences.

**Video Grounding.** Video grounding tries to determine the temporal boundaries of the video moment corresponding to the given sentence. Existing supervised methods can be categorized into the top-down and bottom-up frameworks. The top-down approaches pre-define a series of moment proposals and select the target one by multi-modal estimation, including explicit proposals by sliding windows [14, 17, 24, 25] and implicit proposals by multi-granularity anchors after visual-textual interaction [2, 50, 53, 42, 52, 49]. And the bottom-up framework [3, 5, 41] does not pre-define moment proposals and directly predict the probabilities of temporal boundaries across frames. Under the weakly-supervised setting, the MIL-based methods [29, 15, 8] learn the visual-textual alignment by the inter-sample loss. Among them, Mithun et al. [29] utilize text-guided attention to learn the latent alignment between frames and texts. Gao et al. [15] apply an alignment module to learn the visual-textual consistency and devise a detection module to rank moment proposals. And Chen et al. [8] develop a coarse-to-fine manner to detect the accurate moment. Under the reconstruction-based paradigm, Lin et al. [23] rank candidate moment proposals by a language reconstruction reward. And Song et al. [37] further leverage attentional re-construction to rank the proposals.

## 3 Counterfactual Contrastive Learning

### 3.1 The Formulation of Weakly-Supervised Grounding Networks under CCL

Given the sentence query $Q$ and the instance $C$ (i.e. video $V$ or image $I$), we aim to train a network $\text{GNet}(C, Q)$ to detect the most relevant proposal $p$ without any $(p, Q)$ alignment annotations during training, where a proposal means a moment from the video or a region from the image. To illustrate the CCL paradigm clearly, we first formulate the weakly-supervised grounding network $\text{GNet}(C, Q)$ under CCL. Specifically, we decompose $\text{GNet}(C, Q)$ into three important modules:

- Encoder Module $\text{EM}(\cdot)$: It learns the sentence feature $\mathbf{q}$ and word features $\mathbf{S} = \{\mathbf{s}_n\}_{n=1}^{N}$ from the query $Q$, where $N$ is the word number. It also extracts $T$ proposal features $\mathbf{H} = \{\mathbf{h}_t\}_{t=1}^{T}$ from the instance $C$, i.e. moment or region features.

- Interaction Module $\text{IM}(\cdot)$: It develops the vision-language interaction and output multi-modal proposal features $\mathbf{L} = \{\mathbf{l}_t\}_{t=1}^{T}$. The interaction methods include feature fusion [52], attention-based aggregation [43] and so on.

- Relation Module $\text{RM}(\cdot)$: It builds relation reasoning between proposals and outputs the features $\mathbf{P} = \{\mathbf{p}_t\}_{t=1}^{T}$ and corresponding proposal scores $\mathbf{K} = \{k_t\}_{t=1}^{T}$. Finally, we can obtain the alignment score $\text{Agg}(\mathbf{K})$ for the query-instance sample, where $\text{Agg}(\cdot)$ is an aggregation function based on proposal scores, e.g., averaging the proposal scores, selecting the maximum of these scores or averaging the top-n scores.

### 3.2 The CCL Architecture

In this section, we introduce our CCL paradigm in detail. As shown in Figure 2, we first use the network $\text{GNet}(C, Q)$ to estimate the given query-instance sample. We then apply the gradient-based method to select the critical and inessential proposals. We then construct robust counterfactual transformations (RCT) based on the inessential proposal set and design destructive counterfactual transformations (DCT) according to the critical proposal set. After it, the network $\text{GNet}(C, Q)$ with RCT and DCT generates counterfactual results for the query-instance sample, including positive and negative results corresponding to RCT and DCT, respectively. Finally, we can develop sufficient counterfactual contrastive training with ranking and consistency losses.

Concretely, we first pre-train $\text{GNet}(C, Q)$ with a conventional MIL-based paradigm. For a sample $(C, Q)$, we introduce the randomly-selected instance $\overline{C}$ and query $\overline{Q}$ to construct the negative samples $(\overline{C}, Q)$ and $(C, \overline{Q})$. We then calculate the alignment score $\text{Agg}(\mathbf{K})$ for $(C, Q)$, and compute $\text{Agg}(\mathbf{K}_{\overline{c}})$

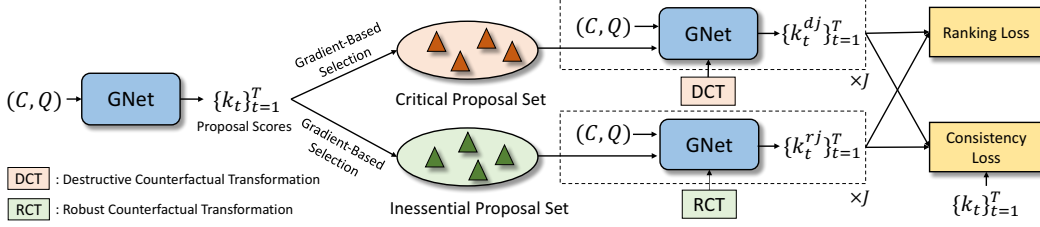

Figure 2: The framework of counterfactual contrastive learning for WSVLG.

and $\mathrm{Agg}(\mathbf{K}_{\overline{q}})$ for $(\overline{C}, Q)$ and $(C, \overline{Q})$. The MIL-based triplet loss $\mathcal{L}_{tri}$ is given by

$$\mathcal{L}_{tri} = \max(0, \ \Delta_{mil} - \mathrm{Agg}(\mathbf{K}) + \mathrm{Agg}(\mathbf{K}_{\overline{c}})) + \max(0, \ \Delta_{mil} - \mathrm{Agg}(\mathbf{K}) + \mathrm{Agg}(\mathbf{K}_{\overline{q}})), \quad (1)$$

where $\Delta_{mil}$ is a margin value which is set to 1.0. Besides it, we also add a diversity loss [9] $\mathcal{L}_{div}$ to adjust the score distribution and stabilize the weakly-supervised training, give by

$$\widetilde{k}_t = \frac{\exp(k_t)}{\sum_{t=1}^{T} \exp(k_t)}, \ \ \mathcal{L}_{div} = -\sum_{t=1}^{T} \widetilde{k}_t \log(\widetilde{k}_t), \ \ \mathcal{L}_{mil} = \mathcal{L}_{tri} + \beta \mathcal{L}_{div}, \quad (2)$$

where $\mathcal{L}_{mil}$ is the pre-trained loss and $\beta$ is set to 0.01 to balance two losses.

After MIL-based pretraining, we develop our CCL paradigm to train $\mathrm{GNet}(C, Q)$. The first step is to build the critical proposal set $\mathbf{P}^+$ and inessential proposal set $\mathbf{P}^-$. Specifically, we apply $\mathrm{GNet}(C, Q)$ to generate the proposal features $\mathbf{H}$, proposal scores $\mathbf{K}$ and alignment score $\mathrm{Agg}(\mathbf{K})$ for $(C, Q)$. A simple method can directly select the proposals with higher scores as the critical proposals. But we further employ the gradient-based selection method and apply the modified Grad-CAM [35] to derive the contribution of $t$-th proposal features, given by

$$\mathrm{Contribution}(\mathbf{h}_t) = (\nabla_{\mathbf{h}_t} \mathrm{Agg}(\mathbf{K}))^T \mathbf{1}, \quad (3)$$

where $\mathbf{h}_t$ is the $t$-th proposal features and $\mathbf{1}$ is an all-ones vector. Based on the contribution of each proposal, we can select $M$ most important proposals as $\mathbf{P}^+$. We then randomly choose $M$ proposals from the resting as $\mathbf{P}^-$. With $\mathbf{P}^+/\mathbf{P}^-$, we can construct robust and destructive counterfactual transformations RCT/DCT and apply them to the intermediate process of network inference, which is introduced in the next section. In brief, we apply $\mathrm{GNet}(C, Q)$ with DCT to generate proposal scores $\mathbf{K}^{dj} = \{k_t^{dj}\}_{t=1}^{T}$ as a counterfactual negative result, where we construct $J$ destructive transformations and $\mathbf{K}^{dj}$ is the $j$-th negative result. Likewise, we generate $J$ counterfactual positive results by $\mathrm{GNet}(C, Q)$ with RCT, where $\mathbf{K}^{rj} = \{k_t^{rj}\}_{t=1}^{T}$ is the $j$-th positive result.

Based on the proposal scores $\mathbf{K}$, $\{\mathbf{K}^{rj}\}_{j=1}^{J}$ and $\{\mathbf{K}^{dj}\}_{j=1}^{J}$ of the original, positive and negative results, we can devise the contrastive loss to learn the proposal-language alignment by the score-based and distribution-based difference. On the one hand, the positive results should have higher alignment score than negative results, thus we develop a margin-based ranking loss by

$$\mathcal{L}_{rank} = \max(0, \ \Delta_{rank} - \frac{1}{J} \sum_{j=1}^{J} \mathrm{Agg}(\mathbf{K}^{rj}) + \frac{1}{J} \sum_{j=1}^{J} \mathrm{Agg}(\mathbf{K}^{dj})), \quad (4)$$

where $\Delta_{rank}$ is a margin value which is set to 0.6. On the other hand, we consider the distribution-based consistency loss to maintain the consistency of the score distributions on the original and positive results, and pull the distributions of original and negative results. Concretely, we conduct the softmax with a low temperature on score distributions and then develop the consistency loss by

$$\overline{k}_t^* = \frac{\exp(k_t^*/\tau)}{\sum_{t=1}^{T} \exp(k_t^*/\tau)}, \ \ \mathcal{L}_{cons} = \frac{1}{J} \sum_{j=1}^{J} (-\sum_{t=1}^{T} \overline{k}_t \log(\overline{k}_t^{rj}) + \sum_{t=1}^{T} \overline{k}_t \log(\overline{k}_t^{dj})), \quad (5)$$

where $\tau$ is the softmax temperature, which is set to 0.5 and produces a sharper score distribution over proposals [18]. Here we regard the original normalized results $\{\overline{k}_t\}_{t=1}^{T}$ as the given pseudo labels.

**Training.** We finally combine the two losses to form the counterfactual contrastive loss by

$$\mathcal{L}_{ccl} = \mathcal{L}_{rank} + \lambda \mathcal{L}_{cons}, \quad (6)$$

where we set $\lambda$ to 0.2 for the balance of two losses. During training, we update the network parameters $\Theta$ at every step, so the network $\mathrm{GNet}(C, Q)$ for critical/inessential proposal selection also has the latest parameters instead of retaining the pre-trained parameters.

**Inference.** Our CCL paradigm is only applied to the training process, thus it will not affect the speed of inference. During inference, we can directly input the given sample into the network $\mathrm{GNet}(\hat{C}, \hat{Q})$ and select the proposal $p$ with the highest proposal score $k_t$ as the grounding result.

### 3.3 Counterfactual Transformation

In this section, we introduce three counterfactual transformation strategies that applied to the intermediate process of network inference. Note that because each strategy will affect the subsequent inference, three strategies cannot be used together but can only be applied individually.

**Feature-Level Strategy (FLS).** Given the set $\mathbf{P}^+$ with $M$ critical proposal features $\{\mathbf{h}_t\}_{t=1}^M$, we first design a feature-level DCT to generate counterfactual negative results. Concretely, we damage the features of critical proposals by memory-based replacement. That is, we replace critical proposal features from $\mathrm{EM}(\cdot)$ and input them to $\mathrm{IM}(\cdot)$. A simple method is to replace the features with the all-zero mask vector, but it is easy to distinguish. Further, we maintain a proposal memory bank with $B$ untrainable vectors $\{\mathbf{m}_b^p\}_{b=1}^B$ that have the same dimensionality with $\mathbf{h}_t$. After MIL-based pre-training, we initialize these vectors by the proposal features from randomly-selected different samples. Next, for each feature $\mathbf{h}_t$ from $\mathbf{P}^+$, we calculate the L2-normalized dot-product with $\{\mathbf{m}_b^p\}_{b=1}^B$ and conduct the softmax over these scores as a probability distribution. We next sample a memory vector $\mathbf{m}_i^p$ according to this distribution and replace $\mathbf{h}_t$. After the replacement in a mini-batch, we update these selected memory vectors by $\mathbf{m}_i^p \leftarrow \alpha\mathbf{m}_i^p + (1-\alpha)\frac{1}{|\mathcal{S}_i|}\sum_{t\in\mathcal{S}_i}\mathbf{h}_t$, where $\mathcal{S}_i$ is the set of proposals that are replaced with $\mathbf{m}_i^p$ in the mini-batch and $\alpha$ is the hyper-parameter to control the momentum update. Likewise, the network $\mathrm{GNet}(C, Q)$ with RCT replaces the inessential features in $\mathbf{P}^-$ with the selected memory vectors to produce counterfactual positive results.

**Interaction-Level Strategy (ILS).** The interaction-level strategy is applied to the multi-modal interaction unit $\mathrm{MMI}(\mathbf{h}_t, \mathbf{a}_t)$ in $\mathrm{EM}(\cdot)$, where $\mathbf{a}_t$ is language feature corresponding to the $t$-th proposal feature $\mathbf{h}_t$. Existing grounding approaches [52, 8] often regard the sentence feature $\mathbf{q}$ as $\mathbf{a}_t$. Further, some works [44, 43, 53, 49] apply an attention-based aggregation to dynamically extract language feature $\mathbf{a}_t$ for each proposal from word features $\{\mathbf{s}_n\}_{n=1}^N$. Similar to FLS, we apply a memory-based replacement method. Specifically, we maintain another query memory bank with $B$ untrainable vectors $\{\mathbf{m}_b^q\}_{b=1}^B$, which are initialized by the language features from random samples. For the proposal $\mathbf{h}_t$ in $\mathbf{P}^+$, $\mathrm{GNet}(C, Q)$ with DCT replaces the language features $\mathbf{a}_t$ by one selected $\mathbf{m}_i^q$ and generate the counterfactual negative result. The selection and update method is the same as in FLS. On the contrary, we destroy the language features corresponding to the proposal in $\mathbf{P}^-$ while producing counterfactual positive results.

**Relation-Level Strategy (RLS).** Recent grounding approaches [44, 43, 52, 50] often contain a relation module to develop context relations between proposals and we design the RLS to perturb the relation construction. Concretely, we first define the relation edge between proposals as $e_{ij} = (i, j, tp_{ij}, w_{ij})$, which represents an edge from the proposal $j$ to the proposal $i$ with the type $tp_{ij}$ and weight $w_{ij}$. This definition of edges can be widely applied to the directed graph [44], undirected graph [5], pre-built static graph with multi-type edges [43] and learning-based dynamic graph with edge weights [50]. For negative result generation, $\mathrm{GNet}(C, Q)$ with DCT transforms crucial edges $\{(i, j, tp_{ij}, w_{ij}) \mid i \in \mathbf{P}^+\}$ related to proposals in $\mathbf{P}^+$ as following rules: 20% probability of no change, 60% probability of replacing the $j$ with another randomly proposal and 20% probability of changing the edge type $tp_{ij}$ to another randomly type. If the graph only has one-type edges, we set the probability of replacing $j$ to 80%. Note that if an edge $e_{ij}$ is undirected, the transformation of $e_{ij}$ will also affect the reversed $e_{ji}$. Thus, during positive result generation, we only transform the edges $\{(i, j, tp_{ij}, w_{ij}) \mid i \in \mathbf{P}^- \text{and } j \notin \mathbf{P}^+\}$ if the relation graph is undirected.

### 3.4 Concrete Grounding Networks Under CCL

In this section, we introduce concrete grounding networks to verify our CCL paradigm, where we adopt the simple and mature components rather than complex designs as in [44, 43, 50, 5]. We briefly describe these components and the details are introduced in Section 1 of the supplementary material.

**Video Grounding Network (VGN).** In $\text{EM}(\cdot)$, we apply a Bi-GRU [10] to model the pre-extracted visual features and obtain contextual features $\{\mathbf{f}_i\}_{i=1}^F$. We define an anchor with the boundaries (s, e) as a moment proposal, and its proposal feature is given by $\mathbf{h}_t = \text{MaxPooling}(\{\mathbf{f}_i\}_{i=s}^e)$. We then learn the word features $\{\mathbf{s}_n\}_{n=1}^N$ by another Bi-GRU. In $\text{IM}(\cdot)$, we employ an attention method [3, 49] to aggregate word features for each proposal, and use the gate-based fusion [53] to generate multi-modal features $\{\mathbf{l}_t\}_{t=1}^T$. In $\text{RM}(\cdot)$, we define an undirected static graph $G$ where an edge is built between two proposals if their temporal IoU (i.e. Intersection-Over-Union) is larger than 0.3. Next, we apply the two-layer GAT [39] to capture proposal-proposal relationships and produce the proposal scores $\{k_t\}_{t=1}^T$ by a linear layer. Finally, we select a top-n aggregation function $\text{Agg}(\cdot)$ to compute the alignment score by $\text{Agg}(\{k_t\}_{t=1}^T) = \frac{1}{|\mathcal{S}_{top}|}\sum_{t \in \mathcal{S}_{top}} k_t$, where $\mathcal{S}_{top}$ is the set of top-n scores.

**Image Grounding Network (IGN).** In $\text{EM}(\cdot)$, we use the pre-trained Faster R-CNN [32] to extract visual features $\mathbf{h}_t$ and spatial features $\mathbf{h}_t^s = [x_t^s, y_t^s, w_t^s, h_t^s]$ for each proposal, where $(x_t^s, y_t^s)$ are the normalized center coordinates of the proposal and $(w_t^s, h_t^s)$ are the normalized width and height. Likewise, we apply a Bi-GRU to learn the word features $\{\mathbf{s}_n\}_{n=1}^N$. In $\text{IM}(\cdot)$, we adopt the proposal-word attention [44, 43] to learn proposal-aware language feature $\mathbf{c}_t$ for each $\mathbf{h}_t$, and simply fuse them by $\mathbf{l}_t = \mathbf{W}^l[\mathbf{h}_t; \mathbf{c}_t] + \mathbf{b}^l$. In $\text{RM}(\cdot)$, we define a directed spatial graph [45, 44, 43] with multi-type edges, where $tp_{ij}$ is based on the spatial features $\mathbf{h}_i^s$ and $\mathbf{h}_j^s$. The details are introduced in Section 1.2 of the supplementary material. Next, we apply a GCN with edge-wise gates [22, 45, 43] to model the proposal relations and learn the features $\{\mathbf{x}_t\}_{t=1}^T$. Finally, we fuse them with spatial features by $\mathbf{p}_t = [\mathbf{x}_t; \mathbf{W}^p\mathbf{h}_t^s]$ and calculate the proposal scores $\{k_t\}_{t=1}^T$ by a linear layer with the sigmoid activation. The aggregation function $\text{Agg}(\cdot)$ is consistent with VGN.

## 4 Experiments

### 4.1 Datasets and Evaluation Metrics

We conduct experiments on two large-scale video grounding datasets ActivityCaption [1] and Charades-STA [14], and three large-scale image grounding datasets RefCOCO [47], RefCOCO+ [47] and RefCOCOg [28]. The dataset details are introduced in Section 2 of the supplementary material.

For a fair comparison, we follow previous works [14, 17] to employ the **R@n,IoU=m** as the evaluation metrics for video grounding. Concretely, we first compute the IoU between the predicted moments and ground truth, and **R@n,IoU=m** is the percentage of at least one of the top-n moments having the IoU > m. As for image grounding, we calculate the **Accuracy** as the metric. If the IoU between the selected region and ground truth is larger than 0.5, we regard it as a right grounding result.

### 4.2 Implementation Details

Following previous works [14, 53, 26], we extract C3D [38] features for videos in ActivityCaption and Charades-STA, where the maximum length of feature sequences is 256 and longer sequences are down-sampled. We pre-define moment proposals by sliding windows with widths [16,32,64,96,128,160] and the stride is 1/4 of the window width, where we generate about 140 proposals for each video. For image grounding, we extract 2,048-d visual features and 4-d spatial features for objects by ResNet-101 based Faster R-CNN [32], where we obtain 36 proposals for each image. As for sentences, we extract the 300-d embedding for each word by the pre-trained Glove embedding [31]. As for the hyper-parameters in CCL, we set the proposal number $M$ in $\mathbf{P}^+/\mathbf{P}^-$ to 32 for video grounding and 12 for image grounding. In FLS and ILS, we set the number $B$ of memory vectors to 100, where the coefficient $\alpha$ of the momentum update is set to 0.9. To avoid time-consuming training, the number $J$ of RCT/DCT is set to 3, where each type of transformation strategies is only applied once and produces a counterfactual result. That is, there are 3 counterfactual positive results corresponding to the robust FLS, ILS and RLS, and 3 negative results corresponding to the destructive FLS, ILS and RLS. During MIL-based pretraining, we use an Adam optimizer [13] with the initial learning rate 0.001. We then use another Adam optimizer with the initial learning rate 0.0005 for the CCL training.

### 4.3 Performance Evaluation on Video Grounding

Considering videos contain consecutive and intricate events, video grounding is relatively more difficult than image grounding. So we conduct more experiments on video grounding.

Table 1: Performance comparisons for video grounding on Charades-STA and ActivityCaption. The best results are bold and the results with underlines are the best in baselines. †: MIL-based methods; *: reconstruction-based methods

| Method | Training | Charades-STA | | | | ActivityCaption | | | |
|---|---|---|---|---|---|---|---|---|---|
| | | R@1 IoU=0.5 | R@1 IoU=0.7 | R@5 IoU=0.5 | R@5 IoU=0.7 | R@1 IoU=0.3 | R@1 IoU=0.5 | R@5 IoU=0.3 | R@5 IoU=0.5 |
| CTRL [14] | FS | 23.63 | 8.89 | 58.92 | 29.52 | - | - | - | - |
| TGN [2] | FS | - | - | - | - | 43.81 | 27.93 | 54.56 | 44.20 |
| 2D-TAN [52] | FS | 39.81 | 23.25 | 79.33 | 52.15 | 59.45 | 44.51 | 85.53 | 77.13 |
| WSLLN† [15] | WS | - | - | - | - | 42.80 | 22.70 | - | - |
| TGA† [29] | WS | 19.94 | 8.84 | 65.52 | 33.51 | - | - | - | - |
| CTF† [8] | WS | 27.30 | 12.90 | - | - | 44.30 | 23.60 | - | - |
| SCN* [23] | WS | 23.58 | 9.97 | 71.80 | 38.87 | 47.23 | 29.22 | 71.45 | 55.69 |
| MARN* [37] | WS | 31.94 | 14.81 | 70.00 | 37.40 | 47.01 | 29.95 | 72.02 | 57.49 |
| VGN† | WS | 30.77 | 12.23 | 70.58 | 37.64 | 46.17 | 28.79 | 71.23 | 55.13 |
| VGN+CCL | WS | **33.21** | **15.68** | **73.50** | **41.87** | **50.12** | **31.07** | **77.36** | **61.29** |

Table 2: Ablation results for video grounding about the counterfactual transformations and contrastive loss.

| Setting | Charades-STA | | | | ActivityCaption | | | |
|---|---|---|---|---|---|---|---|---|
| | R@1 IoU=0.5 | R@1 IoU=0.7 | R@5 IoU=0.5 | R@5 IoU=0.7 | R@1 IoU=0.3 | R@1 IoU=0.5 | R@5 IoU=0.3 | R@5 IoU=0.5 |
| VGN (Base) | 30.77 | 12.23 | 70.58 | 37.64 | 46.17 | 28.79 | 71.23 | 55.13 |
| VGN+CCL (Full) | **33.21** | **15.68** | **73.50** | **41.87** | 50.12 | **31.07** | **77.36** | **61.29** |
| VGN+FLS | 32.06 | 14.66 | 72.28 | 39.65 | **50.53** | 30.78 | 75.66 | 59.35 |
| VGN+ILS | 31.60 | 13.78 | 71.76 | 39.15 | 47.68 | 29.32 | 74.37 | 58.16 |
| VGN+RLS | 31.28 | 13.46 | 71.94 | 38.70 | 48.42 | 29.45 | 73.38 | 57.38 |
| w/o. rank loss | 31.16 | 13.22 | 71.56 | 38.29 | 47.19 | 29.83 | 73.14 | 57.20 |
| w/o. cons loss | 32.55 | 14.71 | 72.49 | 40.37 | 48.87 | 29.28 | 76.09 | 59.27 |

**Baseline.** We compare our method with supervised and weakly-supervised methods. Supervised approaches include the early method CTRL [14], TGN with attention-based interaction [2] and 2D-TAN with proposal relation modeling [52]. Weakly-supervised works contain the MIL-based approaches WSLLN [15], TGA [29], CTF [8] and reconstruction-based methods SCN [23], MARN [37].

**Evaluation Results.** Table 1 reports the performance comparison between our method and existing baselines on Charades-STA and ActivityCaption datasets, where **VGN** is the basic model with only MIL-based training and **VGN+CCL** is under our CCL paradigm. Overall, VGN+CCL achieves the best weakly-supervised performance on all criteria of two datasets, while VGN has a close performance to the state-of-the-art baseline MARN. This fact suggests our CCL paradigm can develop sufficient confrontment between counterfactual positive and negative results, and significantly improve the weakly-supervised accuracy. More specifically, reconstruction-based methods SCN and MARN outperform MIL-based approaches WSLLN, TGA, CTF and VGN, which indicates the MIL-based training cannot construct strong supervision signals from randomly-selected negative samples. But our CCL paradigm provides fine-grained contrastive training by counterfactual transformations in the intermediate process of network inference. Moreover, VGN+CCL outperforms the supervised approaches CTRL and TGN and its performance is further close to the state-of-the-art supervised method 2D-TAN. This demonstrates our CCL framework can reduce the gap between supervised and weakly-supervised video grounding methods.

**Ablation Study.** We next perform ablation studies on three counterfactual transformations and the contrastive loss. Concretely, we first train VGN using the CCL paradigm with only one transformation strategy and produce three ablation model **VGN+FLS**, **VGN+ILS** and **VGN+RLS**. For a fair comparison, we keep the number $J$ of positive and negative transformations unchanged, that is, repeat $J$ times using

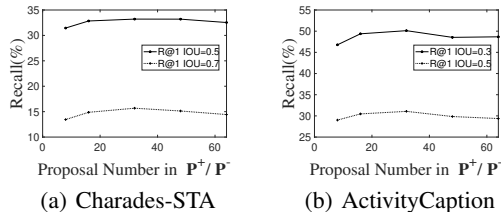

(a) Charades-STA    (b) ActivityCaption

Figure 3: Effect of the number $M$ in $\mathbf{P}^+/\mathbf{P}^-$.

Table 4: Performance comparisons and ablation results for image grounding on RefCOCO, RefCOCO+ and RefCOCOg. The best results are bold and the results with underlines are the best in baselines.

| Method | Settings | RefCOCO | | | RefCOCO+ | | | RefCOCOg |
|---|---|---|---|---|---|---|---|---|
| | | Val | TestA | TestB | Val | TestA | TestB | Val |
| VC | w/o. reg | - | 17.14 | 22.30 | - | 19.74 | 24.05 | 28.14 |
| VC | - | - | 20.91 | 21.77 | - | 25.79 | 25.54 | 33.66 |
| VC | w/o. $\alpha$ | - | 32.68 | 27.22 | - | 34.68 | 28.10 | 29.65 |
| ARN | $L_{lan} + L_{adp}$ | 31.58 | 35.50 | 28.32 | 31.73 | 34.23 | 29.35 | 32.60 |
| ARN | $L_{lan} + L_{adp} + L_{att}$ | 32.17 | 35.35 | 30.28 | 32.78 | 34.35 | 32.13 | 33.09 |
| IGN | Base | 31.05 | 34.39 | 28.16 | 31.13 | 34.44 | 29.59 | 32.17 |
| IGN | CCL | **34.78** | **37.64** | **32.59** | **34.29** | **36.91** | **33.56** | **34.92** |
| IGN | FLS | 33.15 | 36.23 | 31.07 | 32.90 | 35.28 | 32.42 | 33.88 |
| IGN | ILS | 32.28 | 35.27 | 30.50 | 32.13 | 35.74 | 31.74 | 33.23 |
| IGN | RLS | 32.77 | 35.54 | 29.56 | 31.99 | 34.83 | 31.28 | 32.86 |
| IGN | w/o. rank loss | 32.54 | 35.62 | 30.46 | 32.34 | 35.10 | 31.64 | 33.74 |
| IGN | w/o. cons loss | 33.17 | 36.29 | 31.18 | 33.28 | 35.63 | 32.35 | 33.44 |

one strategy. As shown in Table 2, VGN+CCL outperforms three ablation models on almost all metrics of two datasets, but three ablation models achieve the obvious performance improvement than the basic VGN, which illustrates each counterfactual transformation is helpful for weakly-supervised training and the collaboration of three strategies can further enhance the contrastive learning. Next, we discard one loss from the contrastive loss at a time to generate the ablation models **VGN (w/o. rank loss)** and **VGN (w/o. cons loss)**. From the results in Table 2, two ablation models have performance degradation than the full model, indicating each loss is necessary during CCL training. And VGN (w/o. rank loss) achieves the worse accuracy than VGN (w/o. cons loss), demonstrating the importance of the ranking loss.

**Hyper-Parameters Analysis**. We then explore the effect of two crucial hyper-parameter: the proposal number $M$ in $\mathbf{P}^+/\mathbf{P}^-$ and the number $J$ of positive/negative transformations. We first set $M$ to [8, 16, 32, 48, 64] and display the results in Figure 3. We note the model has the best performance on both two datasets while $M$ is set to 32. Because the counterfactual transformation cannot sufficiently destroy crucial grounding clues when $M$ is too small. And the negative results may be easy to distinguish and lead to insufficient contrastive learning while $M$ is too large.

Next, in order to verify whether the larger $J$ can improve model performance, we construct 6 positive/negative counterfactual transformations during CCL training and report the results in Table 3. We can find the model has close performance on both datasets when $J$ is set to 3 and 6, suggesting our CCL paradigm is insensitive to the transformation number.

Table 3: Effect of the number $J$ of positive/negative counterfactual transformations.

| Number | Charades-STA | | ActivityCaption | |
|---|---|---|---|---|
| | R@1 IoU=0.5 | R@1 IoU=0.7 | R@1 IoU=0.3 | R@1 IoU=0.5 |
| 3 | 33.21 | **15.68** | **50.12** | 31.07 |
| 6 | **33.65** | 15.52 | 49.32 | **31.16** |

## 4.4 Performance Evaluation on Image Grounding

**Baseline.** We compare our method with the MIL-based method VC [51] and reconstruction-based approach ARN [26]. Considering the weakly-supervised setting, we use the detected object features from the pre-trained Faster R-CNN rather than the features from ground truth box annotations.

**Evaluation Results.** Table 4 show the evaluation results of our method and existing baselines on three large-scale datasets. The fundamental results are similar to video grounding, that is, the performance of the basic IGN is slightly worse than the existing state-of-the-art method ARN and our CCL paradigm can significantly boost the model accuracy.

**Ablation Study.** We also conduct ablation studies for image grounding about the counterfactual transformations and contrastive loss. From Table 4, we can find the IGN models with FLS, ILS and RLS achieve better performance than the basic one, especially IGN+FLS. This verifies the

Table 5: Performance comparisons with negative sample mining methods.

| Method | Charades-STA | | ActivityCaption | | RefCOCO | RefCOCO+ |
|---|---|---|---|---|---|---|
| | R@1 IoU=0.5 | R@1 IoU=0.7 | R@1 IoU=0.3 | R@1 IoU=0.5 | TestB | TestB |
| Hard Negative Mining | 31.19 | 13.38 | 46.92 | 29.41 | 28.77 | 30.72 |
| Proposal Masking | 31.54 | 14.15 | 47.29 | 29.33 | 30.22 | 31.95 |
| CCL | **33.21** | **15.68** | **50.12** | **31.07** | **32.59** | **33.56** |

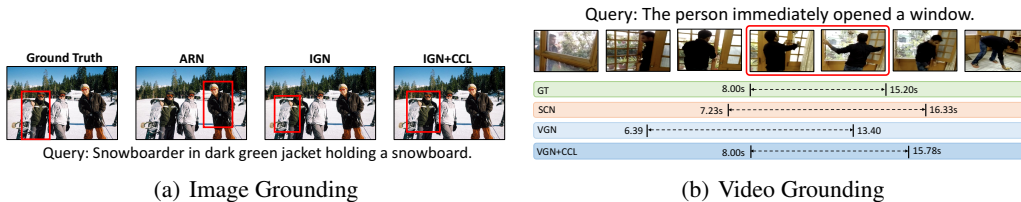

(a) Image Grounding  (b) Video Grounding

Figure 4: Typical examples of weakly-supervised image grounding and video grounding results.

effectiveness of three counterfactual transformations. And similar to video grounding, IGN+CCL with hybrid counterfactual strategies still outperforms all ablation models. From ablation results of the contrastive loss, we observe the performance of IGN (w/o. rank loss) and IGN (w/o. cons loss) drops around 1%~2% on each metric, and IGN (w/o. cons loss) outperforms IGN (w/o. rank loss) on almost all metrics. This fact indicates our CCL training mainly depends on the ranking loss and the consistency loss can further boost the contrastive learning.

### 4.5 Performance Comparison with Negative Sample Mining Approaches

Compared with previous MIL-based approaches [11, 29, 15, 8], our CCL paradigm develops sufficient contrastive training between counterfactual positive and negative results to address the problem that negative samples in MIL-based methods are often easy to distinguish. However, there are other approaches [34, 6] to solve the problem from the perspective of negative sample mining. To further validate the effectiveness of our CCL paradigm, we compare it with two negative sample mining approaches. Concretely, the **Hard Negative Mining** method [34] selects those unmatched vision-language pairs with high alignment scores during model inference as the negative samples for the MIL-based training, which spends much time on negative sample selection. And the **Proposal Masking** method [6] synthesizes hard negative samples by directly masking important proposals and then trains the weakly-supervised model under the MIL-based paradigm. As shown in Table 5, our CCL paradigm achieves better weakly-supervised performance than two stronger baselines from the perspective of negative sample mining. This fact suggests our proposed counterfactual transformations and contrastive training can provide more effective supervision signals for the WSVLG task.

### 4.6 Qualitative Analysis

As shown in Figure 4, we display two typical examples of weakly-supervised grounding results to qualitatively verify the effectiveness of our CCL paradigm. By intuitive comparison, we can find the CCL paradigm can improve the grounding accuracy of the basic IGN and VGN model by distinguishing the target one from plausible proposals, which verifies the effectiveness of our contrastive training between counterfactual results. More examples are shown in Section 3 of the supplementary material.

## 5 Conclusions

In this paper, we propose a novel CCL paradigm for weakly-supervised vision-language grounding. We design the feature-, interaction- and relation-level counterfactual transformations and develop sufficient contrastive training between counterfactual positive and negative results. Extensive experiments on five grounding datasets verify the effectiveness of our CCL paradigm. For future work, we will further explore weakly-supervised contrastive learning and improve training efficiency.

## Broader Impact

This paper introduces a novel CCL paradigm for weakly-supervised vision-language grounding and improves grounding performance. Vision-language grounding is a crucial technique in multi-modal understanding and can be applied to the human-computer interaction field. So this research can promote the development of a multi-modal interaction system and facilitate people's daily lives. And the exploration of weakly-supervised training in this paper can save the labor cost for data annotations. The failure of this technique may lead to an inaccurate multi-modal understanding and cause the mistake of the system based on the grounding results. Moreover, we validate our method on large-scale public vision-language datasets and do not leverage biases in the data.

## Acknowledgments

This work is supported by the National Key R&D Program of China under Grant No. 2018AAA0100603, Zhejiang Natural Science Foundation LR19F020006 and the National Natural Science Foundation of China under Grant No.61836002, No.U1611461 and No.61751209. This research is supported by the Key Laboratory Foundation of Information Perception and Systems for Public Security of MIIT (Nanjing University of Science and Technology) under Grant 202001 and the Huawei Noah's Ark Lab.

## Footnotes

*Zhou Zhao is the corresponding author.

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
