[Supplementary Material]

# The Supplementary Material: Counterfactual Contrastive Learning for Weakly-Supervised Vision-Language Grounding

**Zhu Zhang**[1,2] , **Zhou Zhao**[1,2]*, **Zhijie Lin**[1] , **Jieming Zhu**[3] , and **Xiuqiang He**[3]

[1]Zhejiang University
[2]Key Laboratory Foundation of Information Perception and Systems for Public Security of MIIT,
Nanjing University of Science and Technology
[3]Huawei Noah's Ark Lab
{zhangzhu, zhaozhou, linzhijie}@zju.edu.cn
{jamie.zhu, hexiuqiang1}@huawei.com

## 1 Details of Grounding Networks

In this section, we introduce the video grounding network VGN and image grounding network IGN in detail, where we mainly adopt some widely-used and mature components.

### 1.1 Video Grounding Network

**Encoder Module.** We first extract the visual features $\{\mathbf{v}_i\}_{i=1}^{F}$ of the given video using a pretrained feature extractor (e.g. 3D-ConvNet [11]). We then apply a Bi-GRU network [3] to learn contextual features $\{\mathbf{f}_i\}_{i=1}^{F}$. Next, we define $T$ moment proposals. Each proposal is defined by the boundaries (s,e) and the proposal feature is given by $\mathbf{h}_t = \mathrm{MaxPooling}(\{\mathbf{f}_i\}_{i=s}^{e})$. For language queries, we first extract the embedding for each word token by a pre-trained Glove embedding [8] and employ another Bi-GRU network to learn word features $\{\mathbf{s}_n\}_{n=1}^{N}$.

**Interaction Module.** Given the proposal features $\{\mathbf{h}_t\}_{t=1}^{T}$ and word features $\{\mathbf{s}_n\}_{n=1}^{N}$, we apply a widely-used cross-modal interaction [18, 2] to incorporate language clues into proposal features. Specifically, we first conduct a proposal-to-word attention to aggregate word features for each proposal by

$$
\begin{aligned}
\gamma_{tn} &= \mathbf{w}_m^{\top}\tanh(\mathbf{W}_1^m\mathbf{h}_t + \mathbf{W}_2^m\mathbf{s}_n + \mathbf{b}^m), \\
\widetilde{\gamma}_{tn} &= \frac{\exp(\gamma_{tn})}{\sum_{n=1}^{N}\exp(\gamma_{tn})}, \ \mathbf{c}_t = \sum_{n=1}^{N}\widetilde{\gamma}_{tn}\mathbf{s}_n,
\end{aligned}
\tag{1}
$$

where $\mathbf{W}_1^m$, $\mathbf{W}_2^m$ are projection matrices, $\mathbf{b}^m$ is the bias and $\mathbf{w}_m^{\top}$ is the row vector. The $\mathbf{c}_t$ is the aggregated language feature relevant to the $t$-th proposal. Next, we employ the gate-based fusion [18] between each pair $(\mathbf{h}_t, \mathbf{c}_t)$ by

$$
\begin{aligned}
\mathbf{g}_t^v &= \sigma(\mathbf{W}^v\mathbf{h}_t + \mathbf{b}^v), \ \mathbf{g}_t^s = \sigma(\mathbf{W}^s\mathbf{c}_t + \mathbf{b}^s), \\
\overline{\mathbf{c}}_t &= \mathbf{c}_t \odot \mathbf{g}_t^v, \ \overline{\mathbf{h}}_t = \mathbf{h}_t \odot \mathbf{g}_t^s,
\end{aligned}
\tag{2}
$$

where $\mathbf{g}_i^v$ is a visual gate and $\mathbf{g}_i^t$ is a textual gate. $\sigma$ is the sigmoid function and $\odot$ means the element-wise multiplication. After it, we obtain the multi-modal features $\{\mathbf{l}_t\}_{t=1}^{T}$ by $\mathbf{l}_t = [\overline{\mathbf{h}}_t; \overline{\mathbf{c}}_t]$.

**Relation Module.** We first define an undirected static graph $G$ on proposals. Specifically, if the temporal IoU of two proposals is larger than 0.3, we connect them with an undirected edge. We then apply the two-layer GAT [12] to capture proposal-proposal relationships by

$$\{\mathbf{p}_t\}_{t=1}^T = \mathrm{GAT}(\{\mathbf{l}_t\}_{t=1}^T, G), \tag{3}$$

where $\{\mathbf{p}_t\}_{t=1}^T$ is the final proposal features. And we next apply a linear layer to estimate the score for each proposal by

$$k_t = \sigma(\mathbf{W}^k \mathbf{p}_t + \mathbf{b}^k), \tag{4}$$

where $\sigma$ is the sigmoid function and we finally select a top-R aggregation function $\mathrm{Agg}(\cdot)$ to compute the alignment score by $\mathrm{Agg}(\{k_t\}_{t=1}^T) = \frac{1}{R}\sum_{t \in \mathcal{S}_{top}} k_t$, where $\mathcal{S}_{top}$ is the set of $R$ highest scores.

**Implementation Details.** We set the dimension of hidden states of each direction in the Bi-GRU network to 128. We set the dimension of almost parameter matrices and bias to 256, including the $\mathbf{W}_1^m$, $\mathbf{W}_2^m$, $\mathbf{b}^m$ in the proposal-to-word attention, $\mathbf{W}^v$, $\mathbf{W}^s$, $\mathbf{b}^v$, $\mathbf{b}^s$ in the gate-based fusion, projection matrices in the GAT layers and $\mathbf{W}^k$, $\mathbf{b}^k$ in the last linear layer. In the aggregation function $\mathrm{Agg}(\cdot)$, we set the number $R$ to 32. If we need select multiple moments during inference, we apply the non-maximum suppression (NMS) with a threshold 0.55.

## 1.2 Image Grounding Network

**Encoder Module.** We first use the pre-trained Faster R-CNN [9] to extract visual features $\mathbf{h}_t$ and spatial features $\mathbf{h}_t^s = [x_t^s, y_t^s, w_t^s, h_t^s]$ of each region proposal, where $(x_t^s, y_t^s)$ are the normalized center coordinates of the proposal and $(w_t^s, h_t^s)$ are the normalized width and height. And we apply a Bi-GRU to learn the word features $\{\mathbf{s}_n\}_{n=1}^N$ based on the word embeddings.

**Interaction Module.** Similar to video grounding, given the proposal features $\{\mathbf{h}_t\}_{t=1}^T$ and word features $\{\mathbf{s}_n\}_{n=1}^N$, we first conduct a proposal-to-word attention to aggregate word features by

$$\delta_{tn} = \mathbf{w}_a^\top \tanh(\mathbf{W}_1^a \mathbf{h}_t + \mathbf{W}_2^a \mathbf{s}_n + \mathbf{b}^a),$$
$$\widetilde{\delta}_{tn} = \frac{\exp(\delta_{tn})}{\sum_{n=1}^N \exp(\delta_{tn})}, \ \mathbf{c}_t = \sum_{n=1}^N \widetilde{\delta}_{tn}\mathbf{s}_n, \tag{5}$$

where $\mathbf{c}_t$ is the aggregated language feature relevant to the $t$-th region proposal. Next, we simply fuse them by $\mathbf{l}_t = \mathbf{W}^l[\mathbf{h}_t; \mathbf{c}_t] + \mathbf{b}^l$.

**Relation Module.** We first define a directed spatial graph with multi-type edges as in [15, 14, 13]. Concretely, given two region proposals $r_i$ and $r_j$ with the normalized location features $[x_i^s, y_i^s, w_i^s, h_i^s]$ and $[x_j^s, y_j^s, w_j^s, h_j^s]$. We can compute the $\mathrm{IoU}_{ij}$ of two proposals, the relative distance $d_{ij}$ between their centers, the relative angle $\theta_{ij} \in [0, 360)$ (i.e. the angle of the vector from the center of proposal $r_i$ to that of proposal $r_j$), and the ratio $\phi_{ij}$ between the relative distance $d_{ij}$ and the diagonal length of the image. We then define the type $tp_{ij}$ of the edge between proposals $r_i$ and $r_j$ as following rules: (1) if $r_i$ completely contains $r_j$, $tp_{ij} = 1$, which means "inside"; (2) if $r_i$ is inside $r_j$, $tp_{ij} = 2$, which means "cover"; (3) if above two cases are false and $\mathrm{IoU}_{ij} > 0.5$, $tp_{ij} = 3$, which means "overlap"; (4) otherwise, when the ratio $\phi_{ij} < 0.5$, $tp_{ij}$ depends on the relative angle $\theta_{ij}$, where $tp_{ij} = \left\lceil \frac{\theta_{ij}}{45} \right\rceil + 3 \in [4, 11]$; and (5) if $\phi_{ij} > 0.5$ and $\mathrm{IoU}_{ij} < 0.5$, there is no edge between two proposal and we set $tp_{ij}$ to 0.

Next, we apply the two-layer GCN with edge-wise gates [5, 15, 13] to model the proposal relations. The first layer of GCN aggregate the context feature along the edges for each proposal, given by

$$\overleftarrow{\mathbf{x}}_i^1 = \sum_{e_{ij}>0} g_{ij}(\overleftarrow{\mathbf{W}}^{r1}\mathbf{l}_j + \mathbf{b}_{ty_{ij}}^{r1}), \ \overrightarrow{\mathbf{x}}_i^1 = \sum_{e_{ji}>0} g_{ij}(\overrightarrow{\mathbf{W}}^{r1}\mathbf{l}_j + \mathbf{b}_{ty_{ij}}^{r1}), \ \overline{\mathbf{x}}_i^1 = \overline{\mathbf{W}}^{r1}\mathbf{l}_j + \overline{\mathbf{b}}^{r1}, \tag{6}$$

where $ty_{ij}$ selects the bias vector for each type of edges, $\overleftarrow{\mathbf{W}}^{r1}$ and $\overrightarrow{\mathbf{W}}^{r1}$ are projection matrices for in-edges and out-edges. The $\overleftarrow{\mathbf{x}}_i^1$ and $\overrightarrow{\mathbf{x}}_i^1$ are the aggregated features from in-edges and out-edges, and $\overline{\mathbf{x}}_i^1$ is the updated feature for itself. The $g_{ij}$ is a scale factor from a edge-wise gate, given by

$$g_{ij} = \sigma(\widetilde{\mathbf{W}}^{r1}\mathbf{l}_j + \widetilde{\mathbf{b}}_{ty_{ij}}^{r1}), \tag{7}$$

Query: The person immediately opened a window.

(a) Charades-STA

Query: The man raises the woman and they turn around and continue dancing.

(b) ActivityCaption

Figure 1: Typical examples of video grounding results from the Ground Truth, SCN, VGN and VGN+CCL models, where we show one example for each dataset.

where $\sigma$ is the sigmoid function and $ty_{ij}$ select the bias vector for each type of edges. We then aggregate these features and obtain the output of the first layer of GCN by

$$\mathbf{x}_i^1 = \text{ReLU}(\overleftarrow{\mathbf{x}}_i^1 + \overrightarrow{\mathbf{x}}_i^1 + \overline{\mathbf{x}}_i^1). \tag{8}$$

Likewise, the second layer of GCN operates on features $\{\mathbf{x}_t^1\}_{t=1}^T$ and we denote the output by $\{\mathbf{x}_t\}_{t=1}^T$, where we omit the superscript for convenience. Next, we fuse proposal features with the spatial features by $\mathbf{p}_t = [\mathbf{x}_t; \mathbf{W}^p \mathbf{h}_t^s]$, where $\mathbf{W}^p$ projects the 4-d $\mathbf{h}_i^s$ to 128-d. Similar to video grounding, we finally apply a linear layer to estimate the score for each proposal based on $\mathbf{p}_t$ and use the same top-R aggregation function to calculate the alignment score.

**Implementation Details.** Likewise, we set the dimension of hidden states of each direction in the Bi-GRU network to 128. And the dimension of almost parameter matrices and bias is set to 256, including the $\mathbf{W}_1^a$, $\mathbf{W}_2^a$, $\mathbf{b}^a$ in the proposal-to-word attention, $\mathbf{W}^l$, $\mathbf{b}^l$ in the fusion layer, matrices and biases in the GCN layer and $\mathbf{W}^p$ in spatial feature projection. In the aggregation function $\text{Agg}(\cdot)$, we set the number $R$ to 12.

## 2 Dataset Details

In this section, we introduce the details of five large-scale vision-language grounding datasets.

**Charades-STA [4]:** The Charades-STA dataset is constructed on the Charades dataset [10], where Gao et al. [4] generate the natural language descriptions for video moments by a semi-automatic method. This dataset includes 9,848 videos about indoor human activities and the average duration of these videos is 29.8s. There are 12,408 query-moment pairs for training and 3,720 for testing.

**ActivityCaption [1]:** This dataset includes 19,209 videos about the complex human activities in daily life and the average duration of these videos is about 2 minutes. Since the testing set is not

Query: Left ambulance.

(a) RefCOCO

Query: Elephant behind the middle one's butt.

(b) RefCOCO+

Query: Snowboarder in dark green jacket holding a snowboard.

(c) RefCOCOg

Figure 2: Typical examples of image grounding results from the Ground Truth, ARN, IGN and IGN+CCL models, where we show one example for each dataset.

public yet, we follow previous works [18, 17] and regard the val_1 set as the validation set and val_2 set as the testing set. There are 37,417, 17,505 and 17,031 sentence-moment pairs for training, validation and testing, respectively. This is the largest video grounding dataset currently.

**RefCOCO [16]:** The RefCOCO dataset includes 142,209 queries for 50,000 objects in 19,994 images from MSCOCO [6]. There are 120,624, 10,834, 5,657 and 5,095 query-object pairs in the training, validation, Test A and Test B set. Each image in RefCOCO contains at least 2 objects of the same type.

**RefCOCO+ [16]:** The RefCOCO+ dataset contains 141,564 queries for 49,856 objects in 19,992 images from MSCOCO [6]. There are 120,191, 10,758, 5,726 and 4,889 query-object pairs in the training, validation, Test A and Test B set. Different from RefCOCO, the RefCOCO+ dataset forbids the absolut location descriptions in queries.

**RefCOCOg [7]:** The RefCOCOg dataset contains 95,010 queries for 49,822 regions in 25,799 images from MSCOCO [6]. There is no public testing set of RefCOCOg. And 80,512 and 4,896 query-object pairs are provided in the training and validation set. Compared with RefCOCO and RefCOCO+, RefCOCOg has the longer referring expression to describe the appearance and location of objects.

## 3 Qualitative Analysis

In this section, we display some examples of grounding results to qualitatively verify the effectiveness of our CCL paradigm, where we show one example for each dataset.

Concretely, as shown in Figure 1, we display two examples for video grounding from Charades-STA and ActivityCaption datasets. By intuitive comparison, we can find the grounding accuracy of the basic VGN is close to that of the baseline SCN, and the VGN+CCL significantly improves model performance. This fact verifies the effectiveness of our contrastive training between counterfactual results. As for image grounding, we show three examples from RefCOCO, RefCOCO+ and Ref-COCOg in Figure 1, where the language query of RefCOCO is relatively simple and the queries from RefCOCO+ and RefCOCOg are longer and more complicated. We can find the image often contains multiple objects with the same type to confuse the grounding network. Similar to video grounding, our CCL paradigm can still boost the grounding accuracy of the basic IGN to distinguish the target one from plausible proposals.

## Footnotes

*Zhou Zhao is the corresponding author.