[Reviews · NeurIPS 2020]

Review 1

Summary and Contributions: A counterfactual contrastive learning paradigm is proposed for weakly-supervised vision-language grounding, which can be regarded as an effective improvement for traditional MIL-based or reconstruction-based WSVLG solutions. Three counterfactual transformation strategies from the feature, interaction and relation-level are designed. Experimental results on five grounding datasets have demonstrated the effectiveness of the proposed method.

Strengths: The proposed contrastive learning paradigm is ingenious and effective for WSVLG. Extensive ablation studies have demonstrated the effectiveness of the proposed method.

Weaknesses: (1) The idea of counterfactual contrastive learning is similar to adversarial erasing in object mining, which has been widely used as an effective strategy in weakly supervised detection[a], semantic segmentation[b] and has also been introduced to vision-language grounding in [c]. However, the authors failed to mention the relation between the two and did not cite related papers. (2) The name of “Relation Module” is not very appropriate, because it includes both relational modeling and score inference. (3) What is the motivation of using gradient-based selection in MIL-based pretraining? What are its advantages compared to the direct selection of the proposals with higher scores as the critical proposals. (4) In Line 203, what kind of component can be called a mature component? Verifying the performance of the algorithm w.r.t simple framework seems unconvincing. (5) For temporally language grounding, the authors should cite and compare with [d], which is quite related and is the state-of-the-art method in this field. [a] K. K. Singh and Y. J. Lee. Hide-and-seek: Forcing a network to be meticulous for weakly-supervised object and action localization. In The IEEE International Conference on Computer Vision (ICCV), 2017 [b] Y. Wei, J. Feng, X. Liang, M.-M. Cheng, Y. Zhao, and S. Yan. Object region mining with adversarial erasing: A simple classification to semantic segmentation approach, CVPR 2017. [c] Liu, Xihui, et al. "Improving referring expression grounding with cross-modal attention-guided erasing." Proceedings of the IEEE Conference on Computer Vision and Pattern Recognition. 2019. [d] Wu, Jie, et al. “Tree-Structured Policy based Progressive Reinforcement Learning for Temporally Language Grounding in Video”, AAAI 2020.

Correctness: The claims and method are correct. It will be more convincing to integrate CCL into the existing SOTA grounding frameworks (e.g., [37][38][44]) and prove its effectiveness.

Clarity: This paper is well-written and easy to follow.

Relation to Prior Work: The idea of counterfactual contrastive learning is similar to adversarial erasing in object mining, which has been widely used as an effective strategy in weakly supervised detection[a], semantic segmentation[b] and has also been introduced to vision-language grounding in [c]. However, the authors failed to mention the relation between the two and did not cite related papers. [a] K. K. Singh and Y. J. Lee. Hide-and-seek: Forcing a network to be meticulous for weakly-supervised object and action localization. In The IEEE International Conference on Computer Vision (ICCV), 2017 [b] Y. Wei, J. Feng, X. Liang, M.-M. Cheng, Y. Zhao, and S. Yan. Object region mining with adversarial erasing: A simple classification to semantic segmentation approach, CVPR 2017. [c] Liu, Xihui, et al. "Improving referring expression grounding with cross-modal attention-guided erasing." Proceedings of the IEEE Conference on Computer Vision and Pattern Recognition. 2019.

Reproducibility: Yes

Additional Feedback: The explanation to the major difference between the proposed method and adversarial erasing mostly addressed my concerns.


Review 2

Summary and Contributions: The paper proposed Counterfactual Contrastive Learning (CCL) for weakly-supervised vision-language grounding. CCL conducts contrastive learning by constructing counterfactual positive/negative samples and produces meaningful alignment score for each proposal, which is different from previous MIL-based and reconstruction methods. Three different types of counterfactual transformation are proposed to facilitate the contrastive learning. Experiments conducted on different vision-language grounding benchmarks demonstrate the effectiveness of CCL.

Strengths: -The method is novel. -Proposed CCL can help model localize the parts of video/image relevant to given query under weak supervision signals. -Experimental results shows that CCL is an effective weakly-supervised vision-language grounding method and CCL outperforms the SOTAs. -Complete ablation study.

Weaknesses: Influences of memory bank size B and the memory update strategy are expected to be discussed.

Correctness: Correct

Clarity: Easy to follow

Relation to Prior Work: Yes. The author dicussed in the paper how the proposed Counterfactual Contrastive Learning works differently from previous MIL-based and reconstruction-based weakly-supervised vision-language grounding methods.

Reproducibility: Yes

Additional Feedback:


Review 3

Summary and Contributions: The paper addresses weakly-supervised vision-language grounding, including video grounding and image grounding. Authors propose counterfactual contrastive learning (CCL) to perform contrastive training between generated counterfactual positive and negative results. The experiments on five vision-language grounding datasets demonstrate the effectiveness of the proposed CCL.

Strengths: + The idea of counterfactual contrastive learning (CCL) is novel and reasonable for weakly-supervised vision-language grounding. + CCL achieves new state-of-the-art performance on five vision-language grounding datasets, which demonstrates its effectiveness. + The ablation study is meaningful and demonstrate the effectiveness of counterfactual transformations and the contrastive loss.

Weaknesses: + Counterfactual Transformation. The generation of counterfactual negative results is reasonable; however, the generation of counterfactual positive results from the inessential proposal set is confusing. How can authors guarantee the positive results have higher alignment scores with the original results than the negative results by using the proposed counterfactual transformation. + The distribution of original, counterfactual negative and counterfactual positive results. In fact, we don’t know what counterfactual positive/negative results are generated in the process of counterfactual transformation. Could you provide some visualization or analysis about the distribution of original, counterfactual negative and positive results?

Correctness: Yes

Clarity: well written

Relation to Prior Work: Yes

Reproducibility: Yes

Additional Feedback: The authors answered my questions and would insert the visualization of distribution of original, counterfactual negative and positive results in the revision. I will keep the original score 6.


Review 4

Summary and Contributions: In this paper, the authors aim at addressing the lack of contrastive problem in weakly supervised vision-language grounding by proposing counterfactual contrastive learning. The proposed CCL generates samples with the proposed counterfactual transformations conducted at the feature-, interaction-, or relation-level. Experiments are conducted on weakly supervised image and video grounding datasets.

Strengths: 1. The proposed methods show good results on both image and video grounding datasets. Ablation studies are sufficient to support the effectiveness of the proposed CCL. 2. To my knowledge, this is the first paper that adopts the recent advances in contrastive learning [6, 15] to the weakly supervised vision-language grounding task. 3. The paper is well written and clear.

Weaknesses: 1. The baseline VGN method, with conventional MIL loss, shows good results that already outperforms the previous MIL-based SOTA CTF [7]. It might be necessary to better present the feature and proposal details in Section 4.2 to help understand the good performance of the baseline. 2. Other than the proposed approach in this paper, many previous MIL studies also try to address the problem that “samples are often easy to distinguish,” mainly from negative sampling mining perspective. The experiment can be strengthened by comparing to the designed stronger baselines, such as selecting samples directly from the proposal set, or with naive hard negative mining. 3. The image grounding part is not evaluated on the (more commonly used) ReferitGame and Flickr30K Entities dataset. [1] S. Kazemzadeh, V. Ordonez, M. Matten, and T. L. Berg. Referit game: Referring to objects in photographs of natural scenes. In EMNLP, 2014. [2] B. A. Plummer, L. Wang, C. M. Cervantes, J. C. Caicedo, J. Hockenmaier, and S. Lazebnik. Flickr30k entities: Collecting region-to-phrase correspondences for richer imageto-sentence models. In IJCV, 2016.

Correctness: The manuscript looks correct.

Clarity: The paper is well written and clear.

Relation to Prior Work: The references are clear.

Reproducibility: Yes

Additional Feedback: ######################## Final rating ######################## Thank you for your feedback. The additional comparisons to "hard negative mining" and "direct proposal masking" well address my previous concern on the comparison to existing "hard negative mining" approaches. Overall, I would like to keep my rating as 7.

[Author Response · NeurIPS 2020]

TableR 1: Performance comparisons about the proposal selection approaches and negative sample mining methods

| Method | Charades-STA | | ActivityCaption | | RefCOCO | | RefCOCO+ | | RefCOCOg |
|---|---|---|---|---|---|---|---|---|---|
| | R@1 IoU=0.5 | R@1 IoU=0.7 | R@1 IoU=0.3 | R@1 IoU=0.5 | TestA | TestB | TestA | TestB | Val |
| score-based selection | 32.65 | 15.24 | 49.30 | 30.77 | 36.41 | 32.27 | 36.48 | 32.60 | 34.54 |
| hard negative mining | 31.19 | 13.38 | 46.92 | 29.41 | 34.74 | 28.77 | 34.69 | 30.72 | 32.66 |
| direct proposal masking | 31.54 | 14.15 | 47.29 | 29.33 | 35.59 | 30.22 | 35.60 | 31.95 | 33.16 |
| full | **33.21** | **15.68** | **50.12** | **31.07** | **37.64** | **32.59** | **36.91** | **33.56** | **34.92** |

# 1 To Reviewer 1

**R1.1 The idea of counterfactual contrastive learning is similar to adversarial erasing [a,b,c].** In fact, the core idea of CCL is completely different from the adversarial erasing works [a,b,c]. The only similarity is the process of detecting the crucial parts by the CAM method [32]. After it, the adversarial erasing approaches hide the crucial parts to make the network further focus on other relevant parts. However, our CCL constructs the counterfactual transformations based on the detected proposals for contrastive training. So the connection between CCL and adversarial erasing is very weak. The key contributions of CCL are the usage of contrastive learning for WSVLG, the transformation strategies from three levels and the design of score-based and distribution-based losses, rather than the conventional gradient-based CAM for proposal selection. We will cite these works and illustrate the differences in revision.

**R1.2 The "relation module" includes relational modeling and score inference.** Thanks for your comments and we will separate the score inference from the relation module as an independent module.

**R1.3 What are its advantages of gradient-based selection compared to the direct selection of the proposals with higher scores as the critical proposals.** As shown in TableR 1, the model with score-based selection achieves a worse performance than the full model with gradient-based selection. By qualitative observation, we find the gradient-based method can select more disperse proposals but the score-based method often chooses proposals with large overlaps.

**R1.4 Verifying the performance of the algorithm w.r.t simple framework seems unconvincing.** Actually, as shown in Tables 1 and 4, our basic VGN and IGN achieve the performance close to SOTA, so they are not simple frameworks and our CCL further improves their accuracy. Moreover, we integrate CCL into the existing SOTA approach CTF [7] to prove its effectiveness, shown in TableR 2.

TableR 2: The CCL paradigm on other networks (R@1).

| Method | Charades-STA | | ActivityCaption | |
|---|---|---|---|---|
| | IoU=0.5 | IoU=0.7 | IoU=0.3 | IoU=0.5 |
| CTF | 27.30 | 12.90 | 44.30 | 23.60 |
| CTF+CCL | **32.11** | **14.57** | **48.61** | **28.73** |

**R1.5 The authors should cite and compare with [d].** Thanks for your suggestion and we will add the missing related works in revision. But [d] is a supervised approach and we mainly compare with weakly-supervised methods.

# 2 To Reviewer 2

**R2.1 Influences of memory bank size B and the memory update strategy are expected to be discussed.** We validate the effect of memory size B and update hyper-parameter $\alpha$. When $\alpha$ is set to 0.9 and B is set to [25, 50, 100, 125, 150], R@1 IoU=0.5 on Charades-STA is [32.86, 32.98, 33.21, 33.13, 33.01]. And when B is set to 100 and $\alpha$ is set to [0.6,0.7,0.8,0.9,1.0], R@1 IoU=0.5 on Charades-STA is [32.89, 33.05, 33.15, 33.21, 33.04]. So our CCL paradigm is relatively robust to B and $\alpha$. More experiment data will be provided in the revision.

# 3 To Reviewer 3

**R3.1 How can authors guarantee the positive results have higher alignment scores than the negative results.** The DCT destroys crucial proposals to generate negative results and the RCT damages inessential ones to generate positive results. Because alignment scores rely on the crucial proposals, it is natural that positive results have higher scores.

**R3.2 Could you provide some visualization or analysis about the distribution of original, counterfactual negative and positive results.** By qualitative observation, we can find the distribution of positive results is close to the original results, but the negative results have an inconsistent distribution with the positive and original ones. Due to the page limitation, we will add the visualization of these distributions in the revision.

# 4 To Reviewer 4

**R4.1 The feature and proposal details of the baseline VGN.** We have presented the details of VGN and IGN in Section 1 of the supplementary material, including the model architecture and parameter settings.

**R4.2 Stronger baselines from negative sample mining perspective.** As shown in TableR 1, we further compare CCL with two baselines, where "hard negative mining" selects those unmatched vision-language pairs with high alignment scores during model inference as the negative samples for MIL-based training, and "direct proposal masking" synthesizes negative samples by directly masking important proposals. Our CCL still outperforms the two baselines.

**R4.3 The image grounding part is not evaluated on the ReferitGame and Flickr30K Entities dataset.** Due to the page limitation, we will add the experiments on ReferitGame and Flickr30k in the revision.

[Meta-Review · NeurIPS 2020]

All reviewers recommend acceptance (to varying degrees) after reviewing the author response. The submission focuses on weakly-supervised vision-language grounding and proposes a novel counterfactual contrastive learning objective. Some initial weaknesses with respect to comparison with hard-negative style approaches have been addressed in the rebuttal. I encourage authors to include these results and other suggested revisions in future versions.